# EFFICIENT VIDEOMAE VIA TEMPORAL PROGRESSIVE TRAINING

## ABSTRACT

Masked autoencoders (MAE) have recently been adapted for video recognition, setting new performance benchmarks. Nonetheless, the computational overhead of training VideoMAE remains a prominent challenge, often demanding extensive GPU resources and days of training. To improve the training efficiency of Video-MAE, this paper presents Temporal Progressive Training (TPT), a simple way to strategically introduce longer video clips along the training process. Specifically, TPT decomposes the intricate task of long-clip reconstruction into a series of step-by-step sub-tasks, progressively transitioning from short video clips to long video clips. Our experiments extensively verify the efficacy and efficiency of TPT. For example, TPT can impressively reduce training costs by factors of $2\times$ on Kinetics-400 and $3\times$ on Something-Something V2, while still matching the performance of VideoMAE. Additionally, TPT consistently shows superior performance than VideoMAE when trained with the same budget.

## 1 INTRODUCTION

Self-supervised pre-training has rapidly emerged as a prevailing paradigm for large-scale representation learning. Among various frameworks, masked autoencoder (MAE) stands as a representative example, enjoying the benefits of simplicity, effectiveness, and robustness across both image and video domains (He et al.; Tong et al., 2022; Feichtenhofer et al., 2022). Yet, the training of MAE models, especially in the context of video recognition, presents a substantial computational challenge, thereby restricting its border access for researchers with limited resources. For instance, training a standard VideoMAE model on Kinetics-400 necessitates a massive investment of up to 5.6 days with the support of 64 GPUs.

Efficiency in learning intrinsically rests on the selection of informative samples and mitigation of redundancy, a principle that have been well-corroborated in prior works in image-based learning (Johnson & Guestrin, 2018; Lin et al., 2017). However, the current design of VideoMAE seems to overlook part of these considerations — instead, VideoMAE directly learns from joint spatiotemporal samples, such as cuboid-shaped data, which carry inherent redundancies. While VideoMAE employs strategies like aggressive masking to expedite the training process, we posit that considerable opportunities remain untapped to further its training speed, particularly by carefully addressing the inefficiencies associated with the temporal dimension.

In this paper, we present *Temporal Progressive Training* (TPT), which can effectively train Video-MAE with much fewer computational overheads by mitigating temporal redundancy. The key idea in TPT is to introduce a strategic "warm-up" philosophy to the intricate task of long-length video reconstruction. As illustrated in Figure 1, TPT breaks down the video reconstruction process into an incremental series of sub-tasks, initiating with short video snippets and gradually extending to full-length clips. Moreover, to calibrate TPT for a balanced interplay between efficiency and efficacy, we undertake an in-depth examination of various facets of this task decomposition strategy, including 1) the specifics of the sub-tasks (*e.g.*, type, amount, order, computation of each sub-task) and 2) the setup of key hyper-parameters (*e.g.*, batch size, learning rate schedule). An intriguing insight from our investigation is that while pinpointing the optimal configurations for TPT demands extra effort, this calibration can be achieved on a relatively small-scale dataset with a limited training budget. More interestingly, this configuration, once identified, can reliably and effectively generalize to a variety of datasets and models.

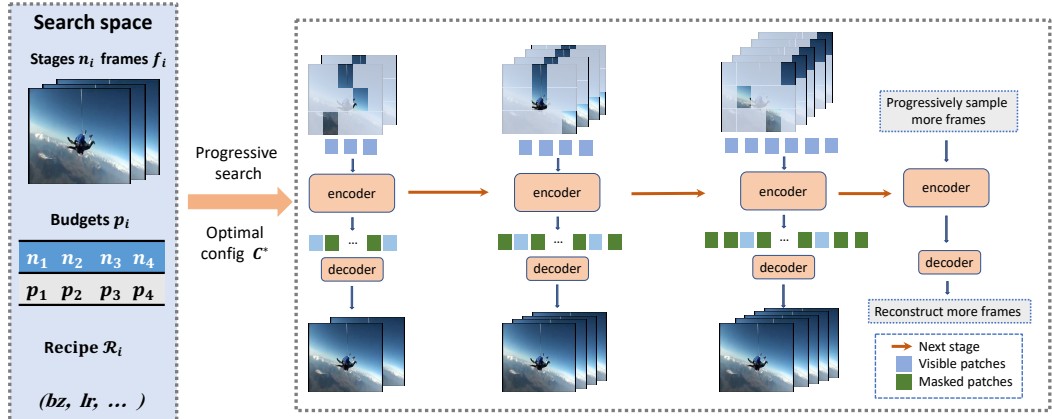

Figure 1: **Overview of a Temporal Progressive Training (TPT) pipeline.** It is a multi-stage Video-MAE training framework. Left: we set up a search space for finding the optimal configuration $\mathcal{C}^*$ of TPT, which is then applied in training (Right). It learns to start with fewer frames for spatial semantics learning and then progressively increases the temporal length to learn at each stage. Finally, the model learns the full spatiotemporal representations.

To comprehensively evaluate the efficiency and efficacy of TPT, we conduct extensive experiments on the Something-Something-V2 (Goyal et al., 2017) and Kinetics-400 (Carreira & Zisserman, 2017) datasets. Our empirical results demonstrate that, when operating within a given training budget, the proposed TPT consistently outperforms VideoMAE (Feichtenhofer et al., 2022; Tong et al., 2022) by a notable margin. Additionally, TPT can competitively match the performance of VideoMAE but at approximately half the training expense for Kinetics-400 and a mere third for Something-Something-V2. We hope this work can serve as a strong benchmark and catalyze further research into efficient video training.

## 2 RELATED WORK

**Masked autoencoder.** The masked autoencoding paradigm has recently gained tremendous success in both natural language processing (Devlin et al., 2018) and computer vision (He et al.). This approach has been used in large language models such as BERT (Devlin et al., 2018) and GPT (Radford et al., 2018; 2019; Brown et al., 2020), which attain excellent representation learning in pre-training. This approach aims to mask a portion of the sentence and then predict the missing context. Driven by the success of NLP, iGPT (Chen et al., 2020) introduces a novel approach that predicts unknown image pixels. This method demonstrates the potential of using a masking-then-predicting pre-training strategy. The original ViT (Dosovitskiy et al., 2021) paper demonstrates that predicting unseen patches in a self-supervised manner can yield good results. Building upon this idea, BEIT (Bao et al., 2021) introduces a method that predicts discrete tokens.

More recently, MAE (He et al.) designs an asymmetric encoder-decoder structure and employs patch masking, significantly reducing computation. Building on this momentum in the image domain, some works further extend MAE to the video domain. BEVT (Wang et al., 2022c) and VIMPAC (Tan et al., 2021) utilize discrete token prediction and demonstrate remarkable performance on action-centric datasets. OmniMAE (Duan et al., 2020) trains a single Vision Transformer on images and videos without labels, achieving comparable or better representations. More recently, VideoMAE (Tong et al., 2022) and ST-MAE (Tong et al., 2022) generalize the ImageMAE (He et al.) approach and show that, even without introducing specific inductive bias or extra labels, vanilla ViT can still generalize well in data-limited scenarios. Our research aligns with these trajectories, but focuses more on improving pre-training efficiency.

**Pre-training and fine-tuning for video classification.** Pre-training followed by fine-tuning is a widely used training strategy for video recognition, where efficient video backbones have been designed by adapting 2D image classification models (He et al., 2016; Simonyan & Zisserman, 2015; Szegedy et al., 2015). Researchers have added video-specific designs into 2D blocks (Liu et al.,

2020; Li et al., 2020b; Lin et al., 2019; Jiang et al., 2019; Li et al., 2021; Sudhakaran et al., 2020; Li et al., 2020a) or incorporated 3D or attention-style blocks (Wang et al., 2018; Wang & Gupta, 2018; Zhou et al., 2018; Chen et al., 2019; Cao et al., 2019; Yin et al., 2020) to combine the benefits of image pre-trained models and domain knowledge. Research on designing efficient 3D CNN backbones has also gained attention (Carreira & Zisserman, 2017; Kondratyuk et al., 2021; Feichtenhofer, 2020; Feichtenhofer et al., 2019; Tran et al., 2018), and recent studies (Li et al., 2022a) have demonstrated that 3D CNNs can also benefit from image pre-training. Recently, ViT (Dosovitskiy et al., 2021) has started to make significant strides in many tasks, including video understanding. In these transformer-based works (Bertasius et al., 2021; Arnab et al., 2021; Liu et al., 2022; Fan et al., 2021; Bulat et al., 2021; Patrick et al., 2021), large-scale image pre-trained weights are required to avoid overfitting. Some works alternatively suggest utilizing a larger video dataset for pre-training (Ghadiyaram et al., 2019; Miech et al., 2019; Yuan et al., 2021; Zhang et al., 2021; Duan et al., 2020; Wang et al., 2021a), but the introduced computational overheads are substantial. Different from these works, our work uniquely delves into the potential of progressively spatiotemporal learning in a self-supervised pre-training context.

**Spatiotemporal sampling for video.** There is a lot of research focused on designing effective sampling strategies for video understanding, including segments-based sparse sampling (Wang et al., 2016), salience-based clip selection (Korbar et al., 2019), varying input shapes (Wu et al., 2020), and combining spatiotemporal sampling with network design (Feichtenhofer, 2020). Recent works have also explored adaptive sampling methods (Meng et al., 2020; Gowda et al., 2021; Zheng et al., 2020; Zhi et al., 2021; Wang et al., 2021b; 2022d) to improve efficiency and accuracy in video processing. Moreover, ViTs' patchify operation allows for increased flexibility in designing sampling strategies (Wang et al., 2022b; Sharir et al., 2021; Wang & Torresani, 2022). Rather than proposing a novel sampling strategy, our approach focuses on re-evaluating the efficacy of MAE-based self-supervised pre-training from a spatiotemporal sampling perspective.

## 3 METHOD

In this section, we discuss our proposed methods in detail. Specifically, in Section 3.1, we first review ImageMAE (He et al.) and VideoMAE (Tong et al., 2022) as preliminaries. Next, we elaborate on the framework of Temporal Progressive Training (TPT) and present its formulation in Section 3.2. Last, in Section 3.2.1, we explain how to find an optimal training configuration that considers both efficiency and effectiveness.

### 3.1 REVISITING IMAGEMAE AND VIDEOMAE

ImageMAE (He et al.) employs an asymmetric encoder-decoder structure and adopts a masking-then-predicting paradigm for self-supervised image representation learning. It randomly masks out 75% of grid patches for reconstruction. Such a strategy is soon widely adopted due to its simplicity and effectiveness. Later, two concurrent works (Tong et al., 2022; Feichtenhofer et al., 2022) generalize it to the video recognition domain by masking in a video clip with 90% of grid cuboids removed for reconstruction. Both works have achieved remarkable success regarding performance on video recognition benchmarks.

To articulate the training process of VideoMAE using a transformer, consider a raw video $\mathbf{X} \in \mathbb{R}^{C \times T \times H \times W}$ with its spatiotemporal resolution as $T \times H \times W$ and input image channel as $C$. This video first undergoes temporal downsampling of $\boldsymbol{e}X$ with a factor of $\tau$, yielding a low-fps video $\boldsymbol{e}X' \in \mathbb{R}^{C \times T/\tau \times H \times W}$.

Inside the $\boldsymbol{e}X'$, each training step randomly samples $B$ video clips as a training batch. Each clip contains $t$ frames with a spatial resolution of $h \times w$, where $h$ and $w$ are the spatial resolutions after pre-processing of random resizing and cropping. This yields an encoder input $\boldsymbol{e}x$ of VideoMAE with the size of $B \times C \times t \times h \times w$.

Then, the network first does a cube embedding operation with a non-overlapping 3D convolution with the size of $s_t \times s_h \times s_w$, which transforms $\boldsymbol{e}x$ into tokens $\boldsymbol{e}t$ as input of the transformer. After the transformation, $\boldsymbol{e}t$ has the size of $B \times C_p \times (\frac{t}{s_t} \times \frac{h}{s_h} \times \frac{w}{s_w})$, where $C_p$ is the embedding dimension, $s_t, s_h, s_w$ is the temporal and spatial downsampling stride. Here, as in ViT (Dosovitskiy

et al., 2021), we usually have $h = w$ and $s_h = s_w$, therefore, we denote the spatial dimension with $p = \frac{h}{s_h}$ and $t' = \frac{t}{s_t}$ for simplicity.

A subsequent tube mask operation is performed, which first masks a large portion of tokens from $et$ at the spatial dimension of $p \times p$, and then propagates the same mask to other frames across $t'$. After the masking, the encoder will process the sampled visible tokens with the size of $B \times C_p \times (t' \times p \times p \times \rho)$, where $\rho$ is the sampling rate (e.g. 10% in VideoMAE). Then decoder would duplicate a mask token which can fill the unseen token in the corresponding position. So the output of decoder has the same size $B \times C \times t \times h \times w$ as input. At last, a mean squared error (MSE) loss is applied to guide the training, and after training, the learned weights are used as initial for downstream tasks such as video classification in our case. Next, we will discuss how TPT helps accelerate this process.

## 3.2 TEMPORAL PROGRESSIVE TRAINING

As discussed in Section 1, the core concept behind temporal progressive training is to apply a strategic "warm-up" approach to the complex task of reconstructing long-length videos. This "warm-up" strategy entails beginning our training process from a pre-trained checkpoint, demonstrating significant improvements in downstream tasks (He et al., 2019; He et al.; Dosovitskiy et al., 2021). Instead of pre-training on various tasks or datasets, our focus is on integrating this pre-train and fine-tuning concept into VideoMAE training. The key lies in designing a pre-training task for VideoMAE training. We consider the target task of reconstructing long-length videos as our downstream objective, making it feasible to pre-train on shorter videos. Then, we can break down the related tasks into a sequence of progressively shorter video reconstruction tasks. This way, each subsequent task can benefit from pre-training on simpler yet training-efficient tasks. As a result, we not only ease the training difficulty but also reduce the computation simultaneously.

In general, a training configuration in TPT can be defined as $\mathcal{C} = \{f_i, p_i, \mathcal{R}_i\}_i^{n_s}$, where $\mathcal{R}_i = \{bz_i, lr_i\}$. Note there could be more training parameters for $\mathcal{R}_i$, while we found the two are most important for our final performance, and searching for other ones can follow a similar paradigm. Under the definition, the configuration $\mathcal{C}$ of VideoMAE (Tong et al., 2022) can be instantiated as $\{f_0 = 16, p_0 = 1, \{bz_0 = 16, lr_0 = 0.01\}\}$, where all budget is used with 8 group of sampled frames (16 in total), which serves as our baseline. On the other hand, a typical TPT pipeline with $\mathcal{C}$ of multiple stages is illustrated in Figure 1. Next, we will delve into defining sub-task parameters and establishing a practical and universal configuration for training a VideoMAE model.

### 3.2.1 SPECIFICS OF SUB-TASKS

**Type of sub-task.** We define a sub-task as a component of VideoMAE training focused on reconstructing a specific shorter video. Since a video is essentially a sequence of images, our smallest sub-task starts with spatial (image) reconstruction, serving as our initial step. Following spatial pre-training, we have two options: we can directly train on the full-length target video or break it down into more sub-tasks. Firstly, we observe that training on spatial information benefits VideoMAE when allocating the same computational resources. Furthermore, we also explore the order in which sub-tasks should be tackled and find that a progressive manner is more effective than a random one. Thus, our sub-task approach is specifically termed "temporal progressive video reconstruction.

**Number of sub-tasks** $n_s$**.** In our Temporal Progressive Training (TPT) method, we adopt a step-by-step learning approach. We begin by enumerating a parameter called $n_s$ from small to large, starting at 1 and progressing up to the largest stage number. We define the search space for $n_s$ as $\mathbb{S} = 1, 2, 3, 4$.

Additionally, we consider the number of frames, denoted as $f_i$, for each sub-task. We perform conditional sampling for $f_i$ from the set $\mathbb{F} = 2, 4, 8, 12, 16$. The total number of possible stage-frame configurations can be calculated combinatorially as $\sum_{1 < i \leq |\mathbb{F}|} \binom{|\mathbb{F}|}{i}$.

However, we apply heuristics to prune many configurations, reducing the search effort and saving computational resources. In our experiments (as described in Section 5), when the budget is as substantial as that in VideoMAE, we do not search for single-stage configurations with fewer frames.

**Computation budget** $p_i$ **of sub-task.** In our approach, we allocate training resources, denoted as $\beta$, to different training stages. We calculate $\beta$ by multiplying the number of video frames used during training by the computational cost of processing a single frame. To make this allocation more manageable, we divide the budget into predefined portions, such as 0%, 6.25%, 12.5%, 25%, 50%, or 100%, with a total always equaling 100%. We gradually assign these portions to each stage while ensuring we don't exceed the remaining budget. We can convert the allocated budget into the number of training epochs using this formula: (allocated portion for a stage * total frames budget) / (dataset size * frames processed per step). For example, if we allocate 25% of the budget to a stage that processes 2 frames per step, we'll need 1600 epochs for that stage to effectively configure our training. This approach optimizes our training process efficiently.

### 3.2.2 FIND OPTIMAL PROGRESSIVE CONFIGURATION

**Search optimal recipe** $\mathcal{R}_i$**.** For each stage, after the computation budget is fixed, one important issue is making sure the knowledge learned in the previous stage can be smoothly transferred to the next one. Here, two parameters considered to be important to adjust are $bz_i$ and $lr_i$. One reason is that in a stage, the temporal frame number $f_i$ is reduced, and it is obvious we may able to increase the $bz_i$ to reduce the wall-clock GPU hours further.

**Scaling across various budgets.** In our experiments (Section 5), our searched configuration $\mathcal{C}$ on Kinectis-400 and Something-Something V2 datasets, with a 400-epoch pre-training budget ($N_f = 400 \times 16$), outperforms the existing VideoMAE configuration. When we need a configuration with a larger budget (800 or 1600 total epochs), instead of conducting a new search, we scale the existing configuration by a factor of 2 or 4. These scaled configurations also surpass the original VideoMAE configuration, highlighting their effectiveness. TPT is a versatile strategy not tied to specific datasets or model architectures. Configurations obtained through TPT can be transferred across different transformer architectures and model sizes, as we will demonstrate later.

## 4 IMPLEMENTATION

**Datasets.** To validate the effectiveness of our method, we conduct experiments on two large-scale datasets: Kinetics-400 (Carreira & Zisserman, 2017) (K400) and Something-Something V2 (Goyal et al., 2017) (SS-V2). K400 consists of approximately 260K raw videos, each categorized into one of 400 action categories.

For pre-training and fine-tuning on K400, we adopt a fixed-stride sampling strategy, a widely-used approach in the literature (Wang et al., 2018; Feichtenhofer et al., 2019; Feichtenhofer, 2020). SS-V2, on the other hand, comprises about 220K videos showcasing 174 predefined human-object interactions with everyday objects. To handle the short videos (i.e., 3 seconds per video) in SS-V2, we employ a segments-based sampling method (Wang et al., 2016). To ensure fair comparisons between different experiments, we set the spatial resolution of all input frames to $224 \times 224$ by default. During pre-training, we vary only the temporal dimension $t$.

**Training and testing.** During the *pre-training* stage, we utilize the AdamW optimizer (Loshchilov & Hutter, 2017) with a weight decay of 0.05. To control the learning rate, we adopt a cosine decay learning rate scheduler with an initial learning rate of $1.5e-4$ and warmup epochs of 40, as suggested in (Tong et al., 2022). We perform our proposed search on SS-V2 dataset with a training budget of 400 epochs. Subsequently, we employ the same approach for various models and K400 dataset but scale the training budget linearly to larger values. We use a batch size of 1024 on 64 Tesla-V100 GPUs for all pre-training tasks.

In *fine-tuning,* we adopt a full-model fine-tune scheme to establish fair comparison following (He et al.; Wei et al., 2022; Tong et al., 2022; Feichtenhofer et al., 2022). In *testing,* we use multi-crop multi-view protocol as suggested in (Tong et al., 2022; Feichtenhofer et al., 2022; Wei et al., 2022). For a fair comparison, we use 3 uniformly sampled spatial views for SS-V2 dataset and 5 views for K400 dataset.

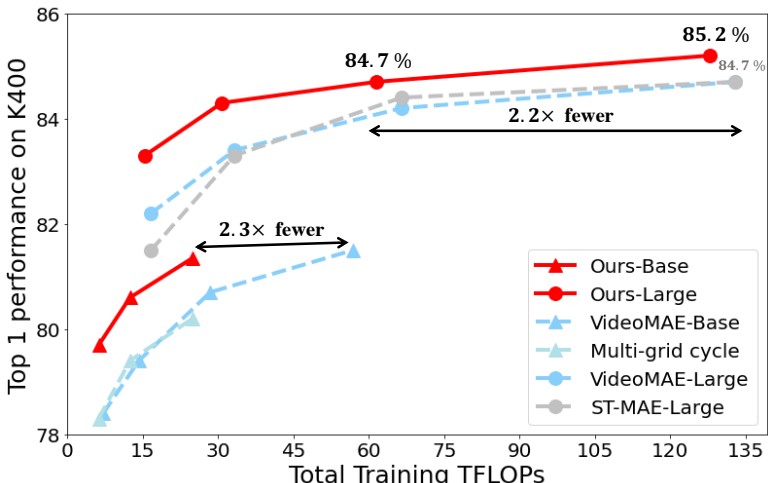

Figure 2: **Accuracy/Training cost trade-off curve on Kinetics-400 dataset (Carreira & Zisserman, 2017) with ViT (Tong et al., 2022) model series.** Each point corresponds to a model with a specific pre-training budget, and we report fully fine-tuning results. Our proposed Temporal Progressive Training framework achieves a preferable trade-off between efficiency and performance.

## 5 EXPERIMENT RESULTS

### 5.1 MAIN RESULTS

**TPT-MAE is efficient.** In this study, we evaluate the efficiency of TPT-MAE with different backbones and training budgets and report the training FLOPs and evaluation top1 accuracy in Figure 2. Our proposed approach consistently reduces model training computation while maintaining uncompromised performance. For instance, using the 1600-epoch training scheduler, TPT-MAE reduces VideoMAE training epochs by $2\times$ and FLOPs by $54\%$, demonstrating superior performance and efficiency trade-off. Moreover, TPT-MAE consistently reduces training epochs by $2\times$ with reduced training budgets while maintaining up to $7.5\%$ reduction in training FLOPs compared to VideoMAE. The results suggest that TPT has a more significant impact on larger models with longer training schedules.

**TPT-MAE is effective.** We further compare the TPT-MAE with other MAE models at comparable FLOPs. Our experimental results demonstrate that TPT-MAE outperforms other MAE models at different training budgets, providing a $0.5\%$ improvement over the ViT-L model using the 1600-epoch setting. These findings showcase the effectiveness of the proposed TPT, which also generalizes well to various computational budgets and application scenarios.

### 5.2 SoTA COMPARISON

To assess the effectiveness of our proposed approach, we compare the performance and efficiency of our VideoMAE model with the state-of-the-art (SoTA) results reported in the literature.

**Kinetics-400.** We first pre-train the TPT-MAE on K-400 dataset for 1600 epochs for a fair comparison. The TPT-MAE consistently improves performance of VideoMAE: $+0.5\%$ up on VideoMAE-Large, and $+0.4\%$ on VideoMAE-Huge. This shows the proposed TPT can learn better spatiotemporal semantics.

**Something-something V2.** Our experiments on SS-V2 dataset reveal similar patterns, as TPT-MAE consistently enhances the performance of VideoMAE. Specifically, we observe performance improvements of $+0.2\%$ on VideoMAE-Large and $+0.6\%$ on ST-MAE-Huge. The consistent results on both large-scale datasets indicate that the proposed approach generalizes well to datasets of varying sizes and models.

| Training Method | Backbone | Pre-train on | param. | GFLOPs | input size | SSV2 | K400 |
|---|---|---|---|---|---|---|---|
| supervised | Slowfast (Feichtenhofer et al., 2019) | K400 | 53M | 106 | $32\times224^2$ | 63.1 | 79.8 |
| supervised | MViT-B (Fan et al., 2021) | K400 | 37M | 455 | $64\times224^2$ | 67.7 | 81.2 |
| supervised | MViTv2-B (Li et al., 2022b) | K400 | 51M | 225 | $32\times224^2$ | 70.5 | 82.9 |
| supervised | MotionFormer (Patrick et al., 2021) | IN-21K+K400 | 109M | 370 | $32\times224^2$ | 66.5 | 80.2 |
| supervised | Swin-B (Liu et al., 2022) | IN-21K+K400 | 88M | 321 | $32\times224^2$ | 69.6 | 81.1 |
| supervised | TimeSformer (Bertasius et al., 2021) | IN21K | 430M | 5549 | $64\times224^2$ | 62.4 | 80.7 |
| MaskFeat (Wei et al., 2022) | MViTv2-L (Li et al., 2022b) | K400 | 218M | 2828 | $40\times224^2$ | 74.4 | 84.3 |
| VideoMAE (Tong et al., 2022) | ViT-L | K400 | 304M | 598 | $16\times224^2$ | 74.0 | 84.7 |
| ST-MAE (Feichtenhofer et al., 2022) | ViT-L | K400 | 304M | 598 | $16\times224^2$ | 72.1 | 84.8 |
| TPT-MAE | ViT-L | K400 | 304M | 598 | $16\times224^2$ | **74.2** | **85.2** |
| ST-MAE (Feichtenhofer et al., 2022) | ViT-H | K400 | 632M | 1193 | $16\times224^2$ | 74.1 | 85.1 |
| TPT-MAE | ViT-H | K400 | 632M | 1193 | $16\times224^2$ | **74.7** | **85.5** |

Table 1: **Comparisons to previous methods on K400 and SS-V2**. We pre-train all the models for 1600 epochs on K400 and conduct full fine-tuning. On SS-V2, following the ST-MAE (Feichtenhofer et al., 2022), we initialize our model with K400 pre-trained weights without intermediate fine-tuning. We report the Top-1 accuracy on the validation set. Parameters are measured in millions and the input size is frame×spatial resolution×spatial resolution.

| Method | Source data | Target | Top-1 |
|---|---|---|---|
| ImageMAE (He et al.) | ImageNet-1k | pixel | 64.5 |
| VideoMAE (Tong et al., 2022) | SS-V2 | pixel | 67.9 |
| MotionMAE (Yang et al., 2022) | SS-V2 | RGB-diff.+pixel | 68.4 |
| MAM$^2$ (Song et al., 2022) | SS-V2 | RGB-diff.+token | 69.0 |
| M$^3$Video (Sun et al., 2022) | SS-V2 | trajectory | 69.2 |
| TPT-MAE | SS-V2 | pixel | **69.4** |

Table 2: **Comparison between recent VideoMAE-based methods on SS-V2**. Comparison of TPT-MAE with other SoTA MAE-based methods with a 400-epoch budget and ViT-B model. RGB-diff refers to temporal differences between frames.

**Comparing to recent MAEs.** In our comparisons, TPT-MAE outperforms recent motion autoencoder (MAE) approaches on video data (Table 2). This highlights TPT's effectiveness in spatiotemporal learning. Additionally, we demonstrate TPT-MAE's versatility by applying it to various backbone architectures (Table 3). Compared to supervised pre-training and VideoMAE training, TPT-MAE consistently performs better, owing to its progressive training approach, which enhances the model's grasp of temporal dynamics while retaining strong spatial awareness.

## 5.3 Ablations

We conduct ablations on SS-V2. We use ViT-B as the backbone and 400-epoch budgets, then fine-tuned for 30 epochs unless specified. We use the same training and fine-tuning configuration as mentioned in Section 3.

**Number of sub-tasks.** Table 4 studies the impact of different training stages. In this ablation, we uniformly distribute the training resources to each stage. An early stage takes half of the frames

| Model | Method | Top-1 |
|---|---|---|
| VideoMAE-16f (Tong et al., 2022) | ImageNet-21k sup. | 61.8 |
| | VideoMAE | 67.6 |
| | TPT-MAE | **69.3** |
| Timesformer-8f (Bertasius et al., 2021) | ImageNet-21k sup. | 59.5 |
| | VideoMAE | 66.3 |
| | TPT-MAE | **67.7** |
| MotionFormer-8f (Patrick et al., 2021) | ImageNet-21k sup. | 66.5 |
| | VideoMAE | 66.2 |
| | TPT-MAE | **67.7** |

Table 3: **Comparison of different backbones on SS-V2.** We compare with VideoMAE under a 400-epoch pre-training computation budget. 8/16f refers to the number of frames.

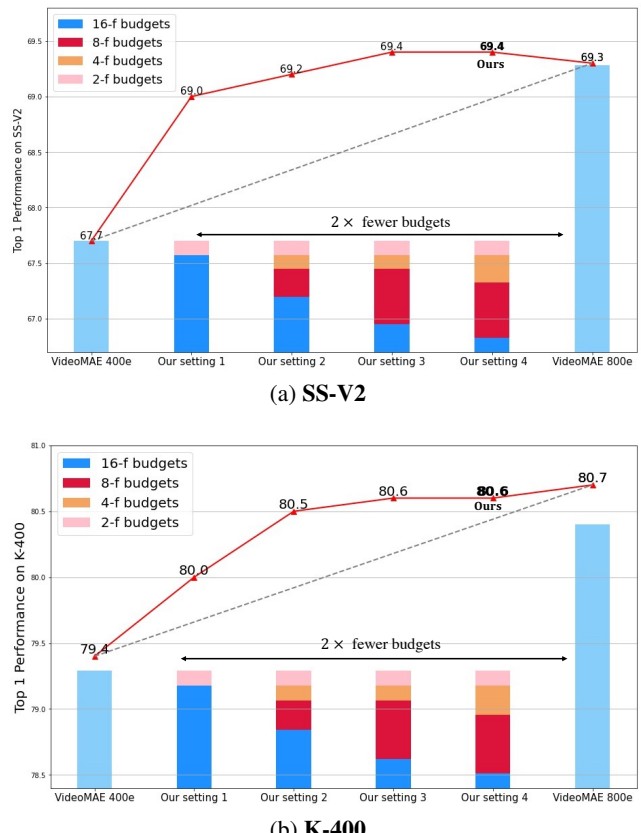

(a) **SS-V2**

(b) **K-400**

Figure 3: Allocate budgets. We use 400-epoch total FLOPs as training budgets for both datasets. Each bar refers to one allocation setting. Each point in the red line represents the Top-1 performance under given training budgets. The first and last bars are our baseline methods with VideoMAE.

| # of sub-tasks | # of frames per task | Top-1 |
|---|---|---|
| 1 | $16\times$ s | 68.4 |
| 2 | $(2, 16)\times$ s | 69.0 |
| 3 | $(2, 4, 16)\times$ s | 69.1 |
| 3 | $(2, 8, 16)\times$ s | 69.2 |
| 3 | $(4, 8, 16)\times$ s | 69.0 |
| 4 | $(2, 4, 8, 16)\times$ s | **69.4** |
| 5 | $(2, 4, 8, 12, 16)\times$ s | 69.1 |

Table 4: Ablation on the number of tasks. 's' denotes our temporal sampling strategy is segments-based (Wang et al., 2016).

| type of sub-task | sampling order | Top-1 |
|---|---|---|
| long-cycle | $(2 \to 4 \to 8 \to 16)$ | **69.4** |
| short-cycle | | 68.8 |
| long-cycle | randomly sampling | 68.9 |
| short-cycle | | 68.4 |

Table 5: Type of sub-task. Upper, for each sub-task, we can choose to use long-cycle or short-cycle style. Bottom, for each sampling, we can fix the order progressively or randomly. Progressive sampling and a single resolution for each stage work the best.

for reconstruction. We empirically find that increasing training stages and with half of the frames improves performance and efficiency. The performance and efficiency gain saturates the 4-stage training scheduler.

**Type of sub-task.** In this study, we explore sub-task types. Initially, we determine the number of sub-tasks to use. Then, we experiment with different sub-task types. First, we examine long-cycle versus short-cycle styles, representing how tasks are divided under the same computation budget. Short-cycle involves repeating predefined sub-tasks ($2 \to 4 \to 8 \to 16$) until the budget is met, while long-cycle assigns a fixed computation for each sub-task. Table 5 reveals that long-cycle performs better. Furthermore, we find that a progressive approach is superior to a random temporal order (e.g., ($4 \to 2 \to 8 \to 16$) ).

| # of learning rate cycles | decay type | Top-1 |
|:---:|:---:|:---:|
| 1 | cosine | 68.7 |
| 2 | cosine | 69.1 |
| 4 | cosine | **69.4** |
| 4 | step | 68.3 |

Table 6: Learning rate scheduler for each sub-task. We ablate the choice of decay strategy and a number of decay cycles. For each stage, a full decay cycle works best, and we choose cosine decay by default.

| 2-*f* | 4-*f* | 8-*f* | 16-*f* | speedup | Top-1 |
|:---:|:---:|:---:|:---:|:---:|:---:|
| 8 | 8 | 8 | 16 | × 1.0 | 69.4 |
| 16 | 16 | 16 | 16 | × 2.0 | **69.4** |
| 32 | 32 | 32 | 16 | × 2.3 | 69.1 |
| 64 | 64 | 64 | 16 | × 2.4 | 68.8 |

Table 7: Batch size for each stage. Different from supervised training (Wu et al., 2020), a larger batch size can degrade the performance in our setting. Hence we choose to keep batch size the same for every stage. Here, we use per-GPU batch size for ablation.

**Allocating computation.** In this study, we examined the trade-off between efficiency and effectiveness with various training schedulers. The results are presented in Figure 3. Compared to the baseline approach (100% training computation on 16-*f*), we observed that assigning more budget to the early stages of training, which involve heavy spatial reconstruction, led to a reduction in training FLOPs and faster training speed. However, this came at the cost of a slight drop in performance. Conversely, allocating the training budget to later stages improved performance at higher training FLOPs.

**Ablation on learning rate scheduler and batch size.** Our investigation into the optimal learning rate and batch size for each training stage is presented in Table 6 and Table 7, respectively. Using a full cosine decay scheduler at each stage produces the best results. We believe this is because such a scheduler helps the model converge more effectively for each stage. Additionally, we conducted experiments on batch sizes and found that scaling the batch size based on the number of frames used for reconstruction hurt performance. This is likely because a larger batch size can cause the model to converge to a local minimum. Based on our empirical results, we selected a consistent batch size of 16 for all stages.

## 5.4 LIMITATION AND FUTURE WORK

**Limitation.** To minimize storage requirements in our experiments, we employ online video decoding. However, when the temporal dimension of the video is short (e.g., two frames), further speedup may be limited by CPU speed. To overcome this bottleneck, one solution is to re-sample the video, reusing decoded frames. For more details, see (Feichtenhofer et al., 2022).

**Future work.** This paper introduces a temporal progressive training framework for VideoMAE. Our framework enables easy extension to the spatial dimension, enhancing its efficiency. Moreover, our proposed method mainly focuses on the data layer. It can potentially generalize to various unsupervised frameworks (Wang et al., 2022a; Ranasinghe et al., 2022; Feichtenhofer et al.) beyond the VideoMAE setting.

## 6 CONCLUSION

We propose a Temporal Progressive Training framework to speed up and enhance VideoMAE. Our framework separates the learning process into low-cost spatial training and more informative-dense but costly temporal training. We begin training with a single frame for spatial semantics learning and progressively increase the temporal clip length at each stage. More importantly, we propose efficiently searching for an optimal temporal progressive training recipe under a small training budget. As a result, under any given training budget, the proposed TPT can outperform (Feichtenhofer et al., 2022; Tong et al., 2022) by a notable margin. In addition, TPT only requires ∼2-3× less training cost to reach the SoTA performance.

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
