# OpenReview forum: "Efficient VideoMAE via Temporal Progressive  Training"
_ICLR.cc/2024/Conference — ICLR 2024 Conference Withdrawn Submission_

### Official Review · Reviewer_eksP · 2023-10-16

**Soundness:** 2 fair
**Presentation:** 2 fair
**Contribution:** 2 fair
**Rating:** 3
**Confidence:** 4

**Summary:**

The authors try to improve the training efficiency of VideoMAE, which adapts the idea of Masked autoencoders (MAE) to video recognition. Specifically, this paper proposes Temporal Progressive Training (TPT), which decomposes the intricate task of long-clip reconstruction into a series of step-by-step sub-tasks, progressively transitioning from short video clips to long video clips. Validation results demonstrate that TPT can improve training efficiency with similar performance.

**Strengths:**

1. Strong performance. Overall, the performance of TPT is good, it can effectively reduce the training costs of VideoMAE by factors of 2 or even 3 without any accuracy drop, which demonstrates the effectiveness of TPT.
2. Empirical validations are extensive. The authors have evaluated TPT on two large-scale video recognition datasets and multiple backbones, and the results on large benchmarks are convincing.

**Weaknesses:**

1. The scope of this paper is not broad enough. The authors only study the training efficiency problem of VideoMAE, which has limited applications in the research community. Although there are papers utilizing the VideoMAE framework, this paper can only benefit those several works in the training speed and cannot generalize to most approaches. Furthermore, a more practical question is whether the training costs really matter for researchers who can afford VideoMAE training.
2. Contribution is limited. The overall idea of this paper, which gradually increases the sampled frames during training, is the same as curriculum learning and there is no other technical contribution in this work.
3. Missing comparison with baseline methods. There is a 'Multi-grid cycle' in Fig. 2 and the reviewer wonders whether this is the re-implementation of Multi-grid training [1]. Apart from that, there are efficient training methods in object recognition [2][3] which should also be re-implemented for comparisons since these methods can be directly adapted to video recognition.

[1] Wu C Y, Girshick R, He K, et al. A multigrid method for efficiently training video models[C]//Proceedings of the IEEE/CVF Conference on Computer Vision and Pattern Recognition. 2020: 153-162.
[2] Wang Y, Yue Y, Lu R, et al. Efficienttrain: Exploring generalized curriculum learning for training visual backbones[C]//Proceedings of the IEEE/CVF International Conference on Computer Vision. 2023: 5852-5864.
[3] Ni Z, Wang Y, Yu J, et al. Deep incubation: Training large models by divide-and-conquering[C]//Proceedings of the IEEE/CVF International Conference on Computer Vision. 2023: 17335-17345.

**Questions:**

1. The reviewer wonders whether the idea of gradually increasing the frames during training could work in vanilla supervised learning in video recognition. If this simple idea works, it can be generalized to all the methods, which should be more exciting than reducing the training time of VideoMAE.
2. There are some typos in this paper: (1) the second paragraph of introduction:  to further its training speed; (2) Sec 3.2.1, paragraph 'Type of sub-task.', last row: ”temporal progressive video reconstruction.

---

### Official Review · Reviewer_G1mA · 2023-11-04

**Soundness:** 2 fair
**Presentation:** 2 fair
**Contribution:** 2 fair
**Rating:** 5
**Confidence:** 5

**Summary:**

This paper introduces a simple method, Temporal Progressive Training (TPT), to improve the training efficiency of VideoMAE pretraining according to a schedule. TPT reduce the computational cost of VideoMAE by gradually introducing longer video clips during training, which divides the whole training period into several sub-tasks. To identify the optimal training configuration, this work conducts a search for the critical parameters within the schedule, shown in the ablation study. The results of experiments demonstrate that TPT significantly reduces training costs while maintaining the performance of VideoMAE.

**Strengths:**

1. This paper propose an efficient method (TPT) for training VideoMAE by compressing input video data in the temporal dimension.
2. This paper tries to explore key parameters within the training schedule through extensive experiments, aiming to find the optimal solution.
3. The proposed method significantly reduces the computational cost while achieving comparable performance with VideoMAE pretraining.

**Weaknesses:**

1. The algorithm and experimental design in this paper is similar to Multigrid[1]. However, this paper primarily focuses on the reduction of temporal length. In Multigrid, reducing spatial size is also an efficient means of reducing computational cost. This paper does not include any experimental analysis of such a design to show the advantages of reducing video temporal length compared to spatial dimension reduction in video pretraining.
2. In Figure 2 of this paper, a comparison is made with the multigrid cycle. However, a clear definition of the multigrid cycle is missing.
3. In 3.2.1, is the calculation of training resources beta reasonable? In the Video Transformer, the computational complexity of the Multi-Head Self-Attention (MHSA) module is O(n^2), where n is the number of input patches. However, in this paper, the calculation resources increase linearly with the length of the video.
4. The writing of the method section is not so clear. It lacks the precise definition of variables and formulas to clarify the algorithm proposed in this paper.
5. In Table 1, the Top-1 accuracy of VideoMAE on K400 is not consistent with its paper reported. The Top-1 accuracy of VideoMAE on K400 is 85.2% rather than 84.7%. Additionally, the results of VideoMAE with ViT-H (86.6% top-1 acc. on K400) is not compared in this paper, which is higher than the results of TPT-MAE with ViT-H. If TPT-MAE is built based on the VideoMAE code repo, the updated baseline results should be compared for fair.
6. For the training efficiency comparison with baselines, This paper only includes FLOPs comparison, without a comparison of Wall-clock GPU hours.
7. In the introduction, it is said that the calibration of optimal configurations can be achieved on a relatively small-scale dataset with a limited training budget, and the configuration can be generalized to other datasets and models. However, it seems that this conclusion is not supported by any experimental results.

[1] Wu, Chao-Yuan, et al. "A multigrid method for efficiently training video models." CVPR 2020.

**Questions:**

I have listed my major concerns in the weaknesses. I hope the authors can provide more rigorous descriptions of some details in the method and reliable experimental results according to the weaknesses.

---

### Official Review · Reviewer_RKG5 · 2023-11-06

**Soundness:** 2 fair
**Presentation:** 1 poor
**Contribution:** 2 fair
**Rating:** 3
**Confidence:** 4

**Summary:**

The paper proposes Temporal Progressive Training (TPT), which accelerates MAE pre-training for videos by optimizing a mix of frame sampling strategies throughout the training process. Compared to VideoMAE, which uses the same architecture and pretraining target as TPT but uses a single sampling strategy, TPT sets new pareto frontier (i.e., higher accuracy at same training budget or same accuracy at reduced training budget) on two popular benchmarks Kinetics-400 and Something-Something-v2. Moreover, the searched plan yields uniformly superior results compared to VideoMAE across different total budgets / datasets / models, avoiding the time-consuming search process for each new task.

**Strengths:**

* The motivation sounds reasonable (it is long known that  and the results look promising on two competitive benchmarks, especially that the searched optimal plan works for both the appearance-focused Kinetics-400 and the motion-focused Something-Something-v2.

* It is interesting that the multi-grid [1] schedule, which is conceptually close to TPT but optimized for supervised training, does not work very well for unsupervised pretraining, potentially making the discovery of this work more valuable.

[1] Chao-Yuan Wu et al., A multigrid method for efficiently training video models, CVPR 2020.

**Weaknesses:**

* **Missing important details about the search method.** Most of the details about the search algorithm are missing. As a result, it is very hard to assess the novelty of the method or the quality of the search result given the current status of the paper. The only clues I can find are vague statements such as '*... we apply heuristics to prune many configurations, reducing the search effort ...*' (In section 3.2.1, what exactly is the heuristics?), '*We graually assign these portions to each stage ...*' (In section 3.2.1, how is the exact move in each step decided?). As the *efficient search* is claimed in the conclusion section, it is expected that either some clear reference to previous works or unambiguous description (e.g., pseudo-code) is provided. The paper also does not provide any intermediate or the final search result for reference.

* **Clarity issues.** Other missing information / self-contradiction / errors are listed as follows, with the more concerning points come first.
  * In Introduction, it is claimed that '*this calibration can be achieved on a relatively small-scale dataset*' which is not well supported by experiments, given that the two datasets SSv2 and K400 are close in size, and it is not mentioned that any subset of SSv2 is used for search.
  * Some confusing statements about experiments: '*using the 1600-epoch training scheduler, TPT-MAE reduces VideoMAE training epochs by 2x and FLOPs by 54%*' (section 5.1, does the 1600-epoch means 1600 passes through all videos, or 1600 passes through number of frames equivalent to an epoch with max numbers per frames? Is this comparing to VideoMAE-3200ep?), '*TPT-MAE consistently reduces training epochs by 2x with reduced training budgets while maintaining up to 7.5% reduction in training FLOPs*' (section 5.1, how is 2x reduction in epochs equivalent to only 7.5% reduction in FLOPs?), '*assigning more budget to the early stages of training ... led to a reduction in training FLOPs and faster training speed*' (section 5.3 the allocating computation part, according to the caption of Figure 3 the bars are actually showing **the same** budget as VideoMAE-400ep, instead of a reduction in training budget), '*came at the cost of a slight drop in performance. Conversely ... at higher training FLOPs*' (section 5.3 the allocating computation part, the conclusion from Figure 3 seems to be the opposite).
  * In Abstract and Introduction, it is claimed that 3x reduction of training cost is achieved for Something-Something-v2, but I cannot find this result in the Experiments section. Instead, figure 3(a) clearly shows '**2x** fewer budgets' on SSv2.
  * Confusing symbols. In section 3.2 paragraph 2, symbols $f_i, p_i, bz_i, lr_i$ come without definition. Symbol $p$ is used as patch size previously in section 3.1.
  * Related Works paragraph 2: It should be *More recently, VideoMAE (Tong et al., 2022) and ST-MAE (**Feichtenhofer et al., 2022**) generalize the ImageMAE ...*

**Questions:**

* In section 3.2.1, what makes it a good approximaton that computational cost is estimated by number of frames, given that the cost of attentions grow quadratically to the number of tokens? Are the training FLOPS numbers in the Experiments section calculated precisely or with this approximation?

* Since the work focuses on the training cost, it would be helpful if the training cost (e.g., wall time / training FLOPS) for the baselines in Table 1 & 2 can also be included for comparison.

* I'm a bit surprised that the mask ratio, which affects the encoder time as much as number of frames, is not included in the search space. Could some explanations be provided about why it is not searched by design?

---

### Official Review · Reviewer_hJSy · 2023-11-10

**Soundness:** 3 good
**Presentation:** 3 good
**Contribution:** 2 fair
**Rating:** 6
**Confidence:** 4

**Summary:**

This work proposes a temporally progressive training scheme, i.e.  progressively transitioning from short video clips to long video over the course of MAE pretraining, resulting in lowering of training costs while retaining performance on K-400 and SS-V2. Results are demonstrated across different model scales and architectures.

**Strengths:**

1. This work revives progressive training for the masked pertaining paradigm and demonstrates it's effectiveness. This a useful contribution as a baseline and also from a practical efficiency perspective and is also simple.

2. Results are demonstrated across different model scales and architectures.

**Weaknesses:**

1. A minor weakness is that technical novelty is somewhat limited with much works on progressive training approaches in the past, though not using masked pretraining paradigm.

2. Details of evaluation are not clear, e.g. how many spatial and temporal crops were used during testing? Implementation details could also be improved, there are not enough details for reproducibility.

3. There is no real exploration of the benefit of TPT from being able to train on longer temporal context. K-400 and SS-V2 are both datasets that are easily solved fairly well with short temporal context. The impact of TPT on tasks beyond classification is also not explored.

4. References could be expanded with discussion on other related work that tries to improves training efficiency in orthogonal ways, e.g. VideoMAEv2, Wang et al, 2023.

**Questions:**

It would be great if the authors could address the discussed weaknesses.